# Risk factors for hypotony after transconjunctival sutureless vitrectomy

Takumi Ando[1]*, Hiroko Terashima[1], Kazuma Fujii[1], Hiromitsu Yoshida[1], Eriko Ueda[1,2], Yohei Nozaki[1], Naoya Shiozaki[1], Kiyoshi Yaoeda[1,3,4], Takeo Fukuchi[1]

1 Department of Ophthalmology, Division of Ophthalmology and Visual Science, Graduate School of Medical and Dental Sciences, Niigata University, Niigata, Japan, 2 Ueda Eye Clinic, Niigata, Japan, 3 Field of Orthoptics and Visual Sciences, Major in Medical and Rehabilitation Sciences, Niigata University of Health and Welfare Graduate School, Niigata, Japan, 4 Yaoeda Eye Clinic, Niigata, Japan

* takumi.ando@med.niigata-u.ac.jp

## Abstract

### Purpose

To identify risk factors for unexpected hypotony after transconjunctival sutureless vitrectomy (TSV).

### Methods

In this retrospective observational study, we defined postoperative hypotony as intra-ocular pressure (IOP) < 6 mmHg on the day after TSV and performed multivariate analysis after dividing patients into hypotony and non-hypotony groups. Peripheral vitrectomy with scleral compression was performed for all patients and completed with normal IOP and no sutures.

### Results

Eight-hundred and forty eyes of 748 consecutive patients who underwent 25-G or 27-G TSV were included. Postoperative hypotony occurred in 139 eyes (16.5%) and was associated with longer axial length (AL) (odds ratio [OR], 0.86; P = 0.001) and no tamponade usage (OR, 0.50; P = 0.001). Postoperative complications occurred more frequently in the hypotony group than in the non-hypotony group (51.1% vs. 11.3%, P < 0.001), especially choroidal fold (47.5%) and hypotony maculopathy (2.2%). On dividing patients without tamponade into 3 AL-based groups, the ≥26-mm group had significantly higher hypotony incidence than the 23–26-mm group (33.3% vs. 18.4%; P = 0.024).

### Conclusion

Longer AL and no tamponade usage influenced hypotony post-TSV. In patients with these factors, especially with AL ≥ 26 mm, surgeons may aggressively consider suturing sclerotomy to minimize hypotony-related complications.

**Data availability statement:** All relevant data are within the paper and its Supporting Information files.

**Funding:** The author(s) received no specific funding for this work.

**Competing interests:** The authors have declared that no competing interests exist.

## Introduction

Transconjunctival sutureless vitrectomy (TSV) is a widely popular surgical procedure that has expanded its application to posterior segment diseases with the evolution of small-gauge instruments such as 25, 23, and 27 gauges [1–3]. Microincision sclerotomy allows for better self-closure and sutureless vitrectomy and has multiple advantages, including less conjunctival scarring, shorter operative time, reduced astigmatism, faster postoperative recovery with less postoperative inflammation, and improved patient comfort [1–5]. Despite these advantages, TSV is associated with postoperative complications such as hypotony, hypotony maculopathy, choroidal detachment, suprachoroidal hemorrhage (SCH), and endophthalmitis, which can cause severe visual impairment [5–10]. In particular, hypotony is a significant trigger for these complications [5–10].

Owing to leakage from sclerotomy, TSV may exhibit a higher hypotony frequency after surgery (3.8%–13.1%) than that of nonsutureless vitrectomy [1–5,11–16]. Preventing hypotony and attaining favorable self-sealing sclerotomy after TSV are topics of much interest for vitreoretinal surgeons to gain the advantages of sutureless vitrectomy. Many methods have been proposed, including the recent technique of scleral needling [15]. However, suturing sclerotomy is often considered if an obvious leak or hypotony occurs at the end of the procedure [15]. Furthermore, unexpected hypotony often occurs the day after surgery, even if the intraocular pressure (IOP) was normal at the end of the surgery [15,16].

Many risk factors have been reported for intraoperative sclerotomy leakage or hypotony after TSV during the transition from 20-gauge to small-gauge vitrectomy, including, for example, younger age, myopia, history of previous vitrectomy, vitreous base dissection, intravitreal triamcinolone, pseudophakic eyes or eyes undergoing combined phacoemulsification and vitrectomy, silicone oil (SO) removal, and the absence of tamponade [14,16–18]. However, few reports have comprehensively examined the risk factors of postoperative hypotony as the primary outcome in large samples. In fact, previous studies included cases in which sclerotomies were sutured and did not examine in detail the risk factors in only sutureless cases. In addition, these reports did not include recent techniques such as 27-G vitrectomy [3] and intrascleral fixation (ISF) [19]. Herein, we examined risk factors for hypotony after TSV in the most recent consecutive patients at our institution.

## Materials and methods

This retrospective observational study adhered to the tenets of the Declaration of Helsinki. Institutional Review Board approval was obtained (approval no. 2023–0166; Niigata University, Niigata, Japan). Written informed consent for surgery was obtained from all patients prior to the procedure. Information about the research was made publicly available on the Niigata University website, and consent was obtained through an opt-out process. Participants were also informed of their right to refuse participation. The ethics committee approved this consent procedure. Data were accessed between October 1 and October 31, 2023. The authors did not have access to any identifiable information during data collection.

## Study participants

Consecutive patients who underwent 25-G or 27-G pars plana vitrectomy (PPV) at Niigata University Medical and Dental Hospital between January 2019 and September 2023 were included. The medical charts of all patients were reviewed. The exclusion criteria were patients (1) with a preoperative IOP of <6 mmHg, (2) whose scleral wounds were sutured at the end of surgery, (3) who underwent glaucoma surgery in the past or simultaneously, (4) with simultaneous peritomy such as scleral buckle placement, and (5) with retinal vitreous disease that could induce hypotony such as retinal detachment with retinectomy, ocular trauma, proliferative vitreoretinopathy, endophthalmitis, and acute retinal necrosis.

## Data collection

The following information was collected from the patients' medical records: age, sex, laterality, diagnosis, preoperative and postoperative IOP, axial length (AL), gauge number (25-G or 27-G), previous vitrectomy, SO removal, triamcinolone acetonide (TA) usage, crystalline lens bag status, endotamponade usage, vitrectomy experience of the surgeon (>4 years or ≤4 years), and surgical time. IOP was measured preoperatively via a noncontact tonometer (TONOREF III; NIDEK, Aichi, Japan) and measured the day after surgery via a Goldmann applanation tonometer. Postoperative hypotony was defined as an IOP of <6 mmHg, as previously reported [14–21]. We divided the patients into hypotony and non-hypotony groups based on their IOP on the first day after surgery.

The AL was measured using the IOL Master optical biometer (Carl Zeiss Meditec AG, Jena, Germany). An A-mode ultrasound device (UD-8000AB; TOMEY, Nagoya, Japan) was used if the intermediate part contained an opacity. Regarding the diagnostic classification, the term "vitreous hemorrhage" includes hemorrhaging in eyes with retinal vein occlusion, posterior vitreous detachment (PVD), and Terson's syndrome, among others, except for proliferative diabetic retinopathy.

The patients were categorized based on the presence or absence of a crystalline lens bag. Patients who had no bag, such as patients with aphakia or who had undergone a simultaneous or previous ISF of intraocular lens (IOL), were included in the one chamber group. Patients who had a bag, such as patients who had undergone lens-sparing vitrectomy or phacoemulsification and IOL implantation simultaneously or previously, were included in the two-chamber group.

## Surgical procedure

Six vitreoretinal surgeons (T.A., H.T., H.Y., E.U., Y.N., and N.S.) performed PPV using the 25-G or 27-G Constellation Vision System (Alcon Laboratories, Fort Worth, TX, USA). The patients were placed under general or local anesthesia. All authors used the RESIGHT 700 Fundus Viewing System (Carl Zeiss Meditec AG). When performing sclerotomy and placing cannulas, the conjunctiva was slightly displaced, and the blade was inserted at an angle of approximately 30° at 3.5–4.0 mm from the corneal limbus. We used the 3-port system in the 25-G and 27-G constellation packs (Constellation Combined Procedure Pak; Alcon Laboratories). If needed, we used chandelier illumination (27-G Oshima Vivid Chandelier; Synergetics, Inc., O'Fallon, MO, USA), depending on the surgeon's preference. After performing a core vitrectomy, PVD, if absent, was induced in all patients using TA (MaQaid; Wakamoto Pharmaceutical, Tokyo, Japan). Thus, the peripheral vitreous skirt was routinely removed with scleral compression, using TA as needed. After performing fluid-gas exchange, we injected air, half air, sulfur hexafluoride, octafluoropropane, and SO, based on the surgeon's judgment. At the end of the surgery, the cannulas were removed, and moderate pressure was applied to the sclerotomy sites with a cotton-tip applicator or forceps. After removing all cannulas, the IOP was checked using tactile examination. If hypotony existed, a balanced salt solution was injected into the anterior chamber of eyes that were not gas-filled, and gas was injected into the vitreous cavity of eyes that were gas-filled to achieve a normal IOP. If scleral wounds were suspected to be poorly self-closing based on the observation of conjunctival bleb formation and hypotony, those scleral wounds were sutured; these cases were excluded, as mentioned previously. The surgery was completed if a normal IOP was confirmed

 

for at least 10 seconds, even if some subconjunctival leakage was observed. At the end of the surgery, 0.1 mL of dexamethasone was injected into the subconjunctiva. The patients' eyes were covered with an eye-patch until the next day.

## Complications

Postoperative complications were defined as observed SCH, choroidal fold, vitreous hemorrhage, hyphema with niveau, and ocular hypertension (≥30 mmHg) on postoperative day 1 and endophthalmitis, hypotony maculopathy, and persistent hypotony up to 1 month. Hypotony maculopathy was defined as transient hypotony wherein the choroidal fold extended into the macula, causing postoperative visual loss and metamorphopsia [6,7]. Vitreous hemorrhage was defined as the situation wherein the optic disc was faintly visible but the retinal vessels were not visible.

## Statistical analysis

We used SPSS, version 25 (IBM, Armonk, NY, USA) for statistical analysis. All data were presented as the mean ± the standard deviation. We used the chi-square test or Fisher's exact test for categorical data and the independent $t$-test for continuous variables to compare the groups with and without postoperative hypotony. Factors that showed strong associations ($P < 0.10$) in univariate analysis were included in the multivariate model of logistic regression to determine the odds ratio (OR) and 95% confidence interval (CI) for the risk of postoperative hypotony. We used area under the receiver operator characteristic curves (AUC) to estimate the predictive accuracy of the multivariable logistic regression models. Statistical significance was set at $P < 0.05$.

## Results

### Patients' characteristics

In total, 840 eyes of 748 patients were included. Postoperative hypotony was observed in 139 (16.5%) eyes on the first day after surgery (Fig 1). Vitrectomy is primarily indicated for patients with a diagnosis of rhegmatogenous retinal detachment or retinal tear, epiretinal membrane, proliferative diabetic retinopathy, and macular hole (>10%), summarized in Table 1. All patients had a mean age of 63.5 ± 12.6 years; 58.7% of patients were men; 52.3% of patients had involvement of the right eye; the mean AL was 24.8 ± 2.0 mm; and 47.7% of patients received tamponade treatment.

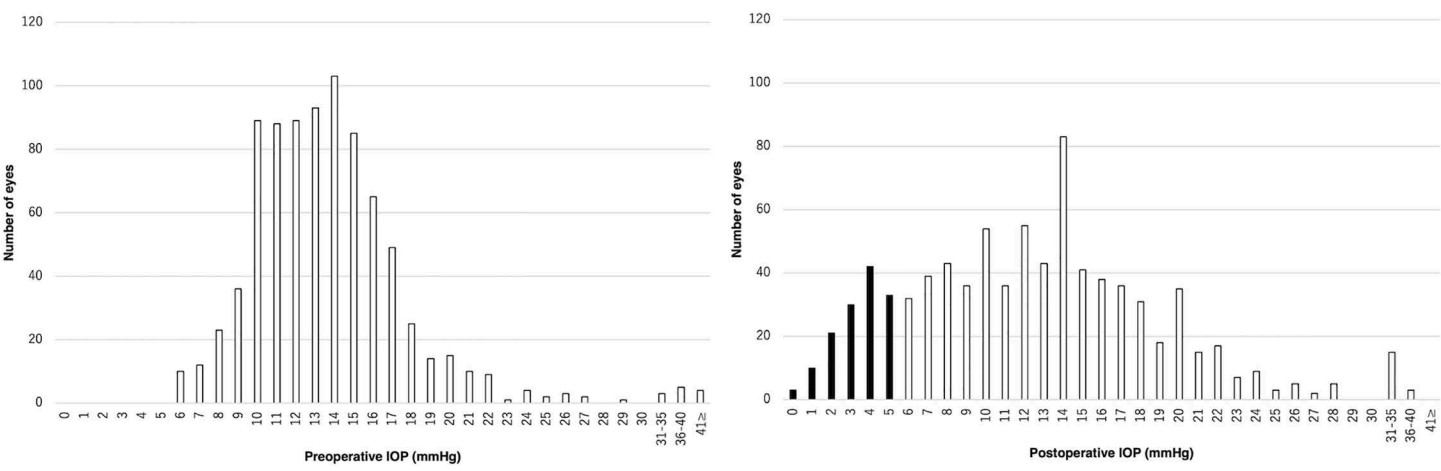

**Fig 1. Histogram of preoperative and postoperative intraocular pressure in all patients.** A total of 840 eyes of 748 patients are included. One hundred thirty-nine (16.5%) eyes experienced postoperative hypotony on the first day after surgery. IOP, intraocular pressure.

Complications occurred in 71/139 (51.1%) eyes in the hypotony group and 79/701 (11.3%) eyes in the non-hypotony group (*P* < 0.001). We encountered no cases of endophthalmitis in both groups, although SCH occurred in six eyes only in the non-hypotony group (*P* = 0.337). Choroidal fold and hypotony maculopathy were significantly observed in the hypotony group (*P* < 0.001 and *P* = 0.004, respectively). All patients with hypotony returned to the pre-surgery IOP level within approximately 1 week; thus, the hypotony was transient (Table 2). Compared to preoperative values, visual acuity deteriorated by ≥2 Snellen lines in two of six eyes with SCH and in two of three eyes with hypotony maculopathy at 3 months postoperatively.

**Table 1. Diagnoses among the patients of this study.**

| Diagnosis | All eyes N (%) | Eyes without tamponade N (%) [a] |
|---|---|---|
| Rhegmatogenous retinal detachment/retinal tear | 180 (21.4) | 3 (1.7) |
| Epiretinal membrane | 162 (19.3) | 141 (87.0) |
| Proliferative diabetic retinopathy | 142 (16.9) | 109 (76.8) |
| Macular hole | 130 (15.5) | 0 (0) |
| Lens and IOL dislocation, aphakia | 76 (9.0) | 70 (92.1) |
| Vitreous hemorrhage | 61 (7.3) | 52 (85.2) |
| Vitreous opacity | 21 (2.5) | 18 (85.7) |
| Silicon oil-filled | 19 (2.3) | 18 (94.7) |
| Vitreomacular traction | 15 (1.8) | 10 (66.7) |
| Subretinal hemorrhage | 14 (1.6) | 11 (78.6) |
| Myopic traction maculopathy | 10 (1.2) | 2 (20.0) |
| Macular hole retinal detachment | 4 (0.5) | 0 (0) |
| Optic disc pit maculopathy | 4 (0.5) | 3 (75.0) |
| Macular edema due to branch or central retinal vein occlusion | 2 (0.2) | 2 (100) |
| Total | 840 (100) | 439 |

IOL, intraocular lens

[a]Percentages indicate the proportion of patients without tamponade for each diagnosis.

**Table 2. Postoperative complications in the hypotony and non-hypotony groups.**

| Observation period | Complication [a] | Hypotony (n = 139) | Non-hypotony (n = 701) | *P* value |
|---|---|---|---|---|
| At postoperative 1 day | Suprachoroidal hemorrhage | 0 (0) | 6 (0.9) | 0.337 |
| | Vitreous hemorrhage [b] | 12 (8.6) | 48 (6.8) | 0.455 |
| | Hyphema with niveau | 3 (2.2) | 17 (2.4) | 0.572 |
| | Choroidal fold | 66 (47.5) | 21 (3.0) | <0.001** |
| | Ocular hypertension (≥30 mmHg) | 0 (0) | 18 (2.6) | 0.037* |
| Up to postoperative 1 month | Endophthalmitis | 0 (0) | 0 (0) | – |
| | Hypotony maculopathy | 3 (2.2) | 0 (0) | 0.004** |
| | Persistent hypotony | 0 (0) | 0 (0) | – |
| | Number of eyes with complications | 71 (51.1) | 79 (11.3) | <0.001** |

*P* < 0.05,

**P* < 0.01

[a]Each complication includes eyes with several complications.

[b]Vitreous hemorrhage was defined as the situation in which the optic disc is faintly visible but the retinal vessels are not visible.

As for intervention, all patients with hypotony were continuously covered with an eye patch until the IOP increased. One patient was observed to have one leaking sclerotomy, which was sutured immediately. One patient was injected with a balanced salt solution into the anterior chamber. Other patients were followed up without contact lenses or pressure patches.

### Risk factors for postoperative hypotony in all patients

Univariate analysis revealed that the following variables were significantly more common in the hypotony group than the non-hypotony group: younger age (60.9 ± 12.5 years vs. 64.0 ± 12.5 years, $P$ = 0.007), longer AL (25.3 ± 2.2 mm vs. 24.7 ± 1.9 mm, $P$ = 0.001), previous vitrectomy (14.4% vs. 7.8%, $P$ = 0.013), SO removal (6.5% vs. 1.4%, $P$ = 0.002), no TA usage (11.5% vs. 5.6%, $P$ = 0.010), and no tamponade usage (64.7% vs. 49.8%, $P$ = 0.002). In multivariate analysis, longer AL (OR: 0.86; 95% CI, 0.78–0.94; $P$ = 0.001) and no tamponade usage (OR: 0.50; 95% CI, 0.33–0.75; $P$ = 0.001) were independently associated with postoperative hypotony (Table 3). The predictive value of this multivariate model using established risk factors including AL and tamponade usage as predictors of hypotony showed low accuracy (AUC = 0.651, 95% CI: 0.602–0.701).

### Risk factors for postoperative hypotony in patients without endotamponade

In 439 eyes of 404 patients without tamponade, univariate analysis revealed that the following factors were significantly more common in the hypotony group than in the non-hypotony group: longer AL (25.0 ± 2.2 vs. 24.3 ± 1.6 mm, $P$ = 0.005), previous vitrectomy (17.8% vs. 10.0%, $P$ = 0.041), and SO removal (8.9% vs. 2.9%, $P$ = 0.017). In multivariate analysis, longer AL (OR: 0.83; 95% CI, 0.73–0.94, $P$ = 0.003) was independently associated with postoperative hypotony (S1 Table).

Based on the AL, we divided the patients without tamponade into three groups: the 20–23 mm group (i.e., AL ≥20 mm but <23 mm), the 23–26 mm group (i.e., AL ≥23 mm but <26 mm), and the ≥26 mm group (i.e., AL ≥26 mm). The incidence of hypotony was 18.2% (n = 16/88) in the 20–23 mm group, 18.4% (n = 53/288) in the 23–26 mm group, and 33.3% (n = 21/63) in the ≥26 mm group. The chi-square test for comparing the frequency of hypotony in each group revealed a significant difference between the 23–26 mm and ≥26 mm groups ($P$ = 0.024) (Fig 2).

## Discussion

In the present study, we investigated the risk factors for hypotony on the first day after TSV, including previously reported factors [13,14,16–18,20]. The major factors were high myopia and no tamponade usage. In the absence of tamponade, hypotony occurred approximately 1.8 times more frequently in myopic patients (i.e., AL ≥26 mm group) than in the other groups. Complications, especially choroidal fold and hypotony maculopathy, were considerably more common in the hypotony group than in the non-hypotony group (51.1% vs. 11.2%). The importance of this study is that it identified risk factors for unexpected postoperative hypotony in patients with normal IOP at the end of TSV. These results may be useful for predicting and preventing postoperative hypotony.

The hypotony rate was higher in this study (16.5%) than in previous reports (3.8%–13.1%) [13–17,20]. We speculate that several reasons might contribute to this finding. First, we included only sutureless cases; a previous study that included only sutureless cases found a high incidence of hypotony (32%) [18]. Second, we routinely removed peripheral vitreous skirts, reducing the plugging effect of the vitreous for sclerotomies [16]. Third, the present study included many high myopic eyes. Woo et al. reported that high myopia is a risk factor for hypotony, with a cutoff AL ≥ 25 mm [16]. When separated by the same cutoff, our study included eyes with longer AL at a higher frequency of 36.7% (309/840) compared to that study (9.7%, 30/308) [16]. Furthermore, the hypotony incidence in the group with AL ≥ 25 mm was 21.7% (67/309), which was similar to the intraoperative sclerotomy suture placement incidence of the group with AL ≥ 25 mm (23.3%,

**Table 3. Demographics of all patients and the analysis findings for risk factors in the postoperative hypotony and non-hypotony groups.**

| | Hypotony (n = 139) | Non-hypotony (n = 701) | Logistic Regression | | | | | |
| | | | Univariate Analysis | | | Multivariate Analysis | | |
| | | | OR | 95% CI | *P* value | OR | 95% CI | *P* value |
|---|---|---|---|---|---|---|---|---|
| **Age, years** | 60.9 ± 12.5 | 64.0 ± 12.5 | 1.02 | 1.01–1.03 | 0.007** | 1.01 | 1.00–1.03 | 0.069 |
| **Sex** | | | | | | | | |
| Male | 76 (54.7) | 417 (59.5) | 0.82 | 0.57–1.19 | 0.293 | | | |
| Female | 63 (45.3) | 284 (40.5) | | | | | | |
| **Laterality** | | | | | | | | |
| Right eye | 78 (56.1) | 361 (51.5) | 1.20 | 0.84–1.74 | 0.319 | | | |
| Left eye | 61 (43.9) | 340 (48.5) | | | | | | |
| **Preoperative IOP, mmHg** | 14.1 ± 5.0 | 13.8 ± 4.8 | 0.99 | 0.95–1.02 | 0.462 | | | |
| **Axial length, mm** | 25.3 ± 2.2 | 24.7 ± 1.9 | 0.88 | 0.80–0.95 | 0.001** | 0.86 | 0.78–0.94 | 0.001** |
| **Gauge number** | | | | | | | | |
| 25 G | 117 (84.2) | 578 (82.5) | 1.13 | 0.69–1.86 | 0.624 | | | |
| 27 G | 22 (15.8) | 123 (17.5) | | | | | | |
| **Previous vitrectomy** | | | | | | | | |
| Yes (≥2 times) | 20 (14.4) | 55 (7.8) | 1.97 | 1.14–3.41 | 0.013* | 1.09 | 0.54–2.22 | 0.815 |
| No (1 time) | 119 (85.6) | 646 (92.2) | | | | | | |
| **Silicone oil removal** | | | | | | | | |
| Yes | 9 (6.5) | 10 (1.4) | 4.78 | 1.91–12.00 | 0.002** | 2.79 | 0.83–9.32 | 0.097 |
| No | 130 (93.5) | 691 (98.6) | | | | | | |
| **TA usage** | | | | | | | | |
| Yes | 123 (88.5) | 662 (94.4) | 0.45 | 0.25–0.84 | 0.010* | 0.86 | 0.38–1.95 | 0.711 |
| No | 16 (11.5) | 39 (5.6) | | | | | | |
| **Lens bag status** | | | | | | | | |
| One chamber | 13 (9.4) | 75 (10.7) | 0.86 | 0.46–1.60 | 0.636 | | | |
| Aphakia | 1 (0.8) | 5 (0.7) | 1.01 | 0.12–8.70 | 0.663 | | | |
| ISF | 12 (8.6) | 70 (10.0) | 0.85 | 0.45–1.62 | 0.624 | | | |
| Two chambers | 126 (90.6) | 626 (89.3) | | | | | | |
| Lens sparing | 16 (11.5) | 48 (6.8) | 1.77 | 0.97–3.22 | 0.058 | | | |
| IOL | 110 (79.1) | 578 (82.5) | 0.81 | 0.51–1.27 | 0.353 | | | |
| **Tamponade usage** | | | | | | | | |
| Yes | 49 (35.3) | 352 (50.2) | 0.55 | 0.37–0.80 | 0.002** | 0.50 | 0.33–0.75 | 0.001** |
| Air | 10 (7.2) | 56 (8.0) | 0.89 | 0.44–1.80 | 0.751 | | | |
| Half air | 2 (1.4) | 28 (4.0) | 0.35 | 0.08–1.49 | 0.102 | | | |
| SF6 | 37 (26.6) | 261 (37.2) | 0.61 | 0.41–0.92 | 0.017* | | | |
| C3F8 | 0 (0.0) | 1 (0.1) | – | – | 0.845 | | | |
| SO | 0 (0.0) | 6 (0.9) | – | – | 0.337 | | | |
| No | 90 (64.7) | 349 (49.8) | | | | | | |
| **Surgeon's vitrectomy experience** | | | | | | | | |
| >4 years | 104 (74.8) | 540 (77.0) | 0.89 | 0.58–1.35 | 0.573 | | | |
| ≤4 years | 35 (25.2) | 161 (23.0) | | | | | | |
| **Surgical time, min** | 48.9 ± 22.0 | 49.9 ± 25.4 | 1.00 | 0.99–1.01 | 0.673 | | | |

*$P < 0.05$, **$P < 0.01$

Note: Data are expressed as the number (%) or as the mean ± the standard deviation. Percentages represent the proportion of patients in the hypotony and non-hypotony groups.

OR, odds ratio; CI, confidence interval; IOP, intraocular pressure; ISF, intrascleral fixation; TA, triamcinolone acetonide; SF6, sulfur hexafluoride; C3F8, octafluoropropane; SO, silicone oil

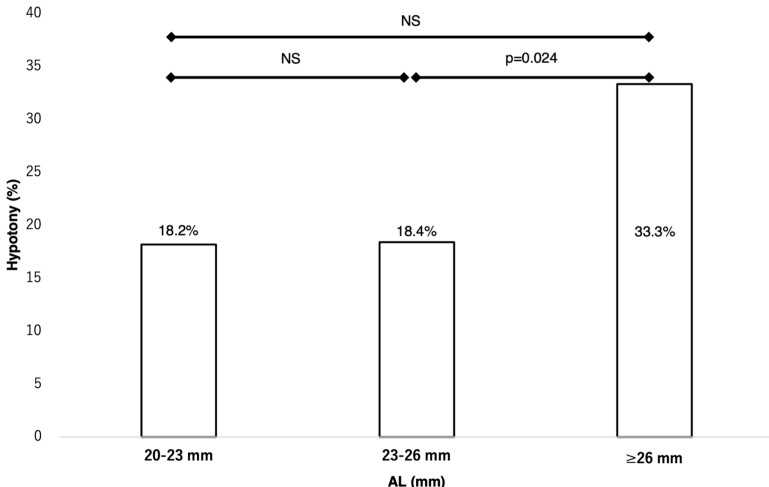

**Fig 2. Comparison of the incidence of hypotony between the axial length groups without tamponade.** The patients without tamponade are divided into three groups, based on the axial length (AL): 20–23 mm, 23–26 mm, and ≥26 mm; the incidence of hypotony is 18.2% (n = 16/88), 18.4% (n = 53/288), and 33.3% (n = 21/63), respectively. The AL ≥26 mm group has a significantly higher frequency of hypotony than the 23–26 mm group; *P* = 0.024. AL, axial length; NS, not significant.

7/30) [16]. Because our procedures were completed without sutures if a normal IOP was maintained, even in cases of subconjunctival leakage, this result implies that intraoperative leakage directly influenced postoperative hypotony in eyes with longer AL. Postoperative blinking may have promoted the leakage of tamponade material through such incomplete sclerotomy [15,22], suggesting that if IOP is normal but leakage is observed in myopic eyes, sclerotomies will not self-close, leading to hypotony.

Our study revealed that high myopia and no tamponade usage were significant factors for hypotony, which occurred frequently in patients with AL ≥ 26 mm without tamponade. AL elongation, younger age, and prior surgery could be associated with scleral rigidity decreasing [16–18,23,24] and sclerotomy leakage [16–18]. Nevertheless, the latter two factors were not significant in our multivariate analysis, which suggested that pathological sclera due to high myopia [25] may play a key role in poor self-closure. Gas tamponade prevents hypotony by its surface tension, promoting the early closure of sclerotomy and preventing leakage postoperatively [26]. However, a recent report on myopic traction maculopathy (MTM) showed that tamponade may have an adverse effect on visual function and retinal morphology [27]. Because high myopia without tamponade could frequently lead to hypotony in cases of TSV, surgeon may consider aggressively suturing sclerotomy in eyes with AL ≥ 26 mm and no tamponade, such as in cases of MTM with pathological sclera.

In addition to high myopia and no tamponade usage, previous vitrectomy and SO removal have been identified as significant risk factors for hypotony following vitrectomy [14,16,17], though this association was only observed in our univariate analysis. If vitreous base shaving is performed, the residual vitreous may become insufficient for wound sealing in subsequent vitrectomy procedures [14,16]. Additionally, reusing the same trocar position may delay the wound healing process due to scleral scarring or thinning [14,16]. In cases of SO removal, the presence of residual SO within the wound may impair adequate self-sealing, and these eyes have typically undergone prior vitrectomy [17]. Although some surgeons in our study opted not to suture in these cases after confirming IOP, there was no deliberate attempt to avoid suturing in all cases. The primary mechanism of hypotony in almost all TSV cases is likely postoperative "silent" wound leakage, except for the ciliary body dysfunction due to uveitis [20]. Given these mechanisms, proactive suturing should still be considered in cases with these risk factors to prevent such leakage, even in modern small-gauge vitrectomy.

An insignificant factor in this study was gauge number, of which 27-TSV was not included in previous reports regarding postoperative hypotony [13–18]. Currently, 27G is the smallest sized instrument and seems to prevent hypotony. This contrary result is similar to that reported in a recent meta-analysis [28], implying that selecting gauge number based on only the prevention of postoperative hypotony might be ineffective.

The anterior segment affects peripheral vitrectomy and greatly relates to postoperative IOP [14,16,17]; however, the lens capsule was an insignificant factor in this study. We speculated that the absence of a lens capsule, which has not been analyzed previously [13–18], may greatly affect hypotony because the patient would be in a state of aphakia, especially before ISF during surgery, and would have better peripheral visibility, and the peripheral vitreous body would be easily removable. We also assumed that postoperative leaks from the main wound or corneal side ports would directly affect the loss of the vitreous cavity. Our results differ from our speculation, indicating that, owing to the improved peripheral visibility of modern wide-viewing systems, significant differences may not exist among various anterior segment statuses in peripheral vitrectomy.

To prevent SCH, avoiding hypotony is crucial, particularly in cases of high myopia. High myopia is associated with a risk of choroidal hemorrhage due to the fragility of the choroidal vasculature as a result of AL elongation [8,9]. In addition, hypotony causes choroidal effusion that stretches and ruptures a long or short posterior ciliary artery [8,29]. SCH after PPV ranges from 0.06% to 4.3% [8,9]. In this study, its frequency was 0.71% (6/840 eyes), which is higher than that (0.06%, 28/48,654 cases) observed in a recent report [30]. All six patients with SCH were in the non-hypotony group and had hyphema and vitreous hemorrhage, including two patients with vision loss. IOP was not measured continuously; however, considering the high rate of hypotony in this study, postoperative hypotony likely induced the high incidence of SCH, which consequently masked their hypotony by hyphema and vitreous hemorrhage.

To our best knowledge, there are few reports on hypotony maculopathy after vitrectomy, and its frequency is unknown [6,7,31]. Hypotony maculopathy after PPV is a rare complication that causes irreversible retinal damage because of the choroidal fold by transient hypotony extending to the macula [6,7]. In contrast, this condition is more commonly associated with complications after glaucoma surgery, with reported incidence rates of 9.4% after trabeculectomy and 6.0% after Ahmed glaucoma valve implantation in eyes with hypotony (IOP < 9 mmHg) [29]. The present study showed the frequency of hypotony maculopathy, which was observed in 2.2% of cases in the hypotony group and 4.5% of cases with choroidal folds. Although infrequent, two of three cases had vision loss. The mechanism of this complication is that the retinal choroidal folds caused by ocular collapse after rapid postoperative leakage from the vitreous cavity induce retinal damage because the vitreous body cannot be expected to provide a cushioning effect. In fact, our three cases did not receive endotamponade, which may suggest that tamponade material prevents the eye globe from collapsing rapidly instead of vitreous body. However, its risk factors should be further investigated in a multicenter study because of the small sample size of the current study.

This study had some limitations. First, our study was retrospective in design and included multiple surgeons. Therefore, we were unable to analyze the relationship between hypotony and subconjunctiva leakage, which could be a significant cause of postoperative hypotony. Second, on the first day after surgery, the IOP was measured approximately 12–24 h postoperatively, which was not a uniform time period. Third, we were unable to show a clear cutoff value for AL using the receiver operating characteristic curve because it was not a useful value. However, we utilized normal practice-based data that were easily comparable to actual clinical practice. Furthermore, among studies with postoperative hypotony as the main outcome, this study comprised the largest number of patients and included recent techniques (27-G vitrectomy and ISF) and only sutureless cases, which we believe reflects relatively recent real-world results.

## Conclusion

A longer AL and no tamponade usage influenced hypotony after TSV. Complications were more common in the hypotony group than in the non-hypotony group, especially choroidal fold and hypotony maculopathy. TSV is relatively safe;

however, it should be noted that there are some complications that can cause vision loss infrequently. For patients with eyes with a longer AL without tamponade, especially AL ≥ 26 mm, surgeons may aggressively consider suturing sclerotomy to minimize hypotony and hypotony-related complications.

## Supporting information

**S1 Table. Demographics of patients without tamponade and the analysis findings for risk factors in the postoperative hypotony and non-hypotony groups.**
(DOCX)

## Acknowledgments

Presented at the 128th Annual Meeting of the Japanese Ophthalmological Society (Tokyo, Japan) on April 18–21, 2024.

## Author contributions

**Conceptualization:** Takumi Ando, Hiroko Terashima.

**Data curation:** Takumi Ando, Hiroko Terashima, Kazuma Fujii.

**Formal analysis:** Takumi Ando, Hiroko Terashima, Kazuma Fujii, Kiyoshi Yaoeda.

**Investigation:** Takumi Ando, Hiroko Terashima, Kazuma Fujii, Kiyoshi Yaoeda.

**Methodology:** Takumi Ando, Hiroko Terashima.

**Project administration:** Takumi Ando, Hiroko Terashima, Takeo Fukuchi.

**Resources:** Takumi Ando, Hiroko Terashima, Kazuma Fujii, Hiromitsu Yoshida, Eriko Ueda, Yohei Nozaki, Naoya Shiozaki.

**Software:** Takumi Ando, Kazuma Fujii.

**Supervision:** Hiroko Terashima, Takeo Fukuchi.

**Validation:** Takumi Ando, Hiroko Terashima, Kazuma Fujii, Kiyoshi Yaoeda.

**Visualization:** Takumi Ando.

**Writing – original draft:** Takumi Ando.

**Writing – review & editing:** Hiroko Terashima, Takeo Fukuchi.

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
