## [Decision Letter · Decision Letter 0]

13 Feb 2025

PONE-D-25-02954Risk factors for hypotony after transconjunctival sutureless vitrectomyPLOS ONE

Dear Dr. Ando,

Thank you for submitting your manuscript to PLOS ONE. After careful consideration, we feel that it has merit but does not fully meet PLOS ONE’s publication criteria as it currently stands. Therefore, we invite you to submit a revised version of the manuscript that addresses the points raised during the review process.

Dear author, your article will be more relevant if findings are compared with intial or prior work on post operative hypotony after transconjunctival vitrectomy. Also include the possible complications of hypotony in your discussion. Kindly revise your article according to the comments and advice of the reviewers. 

We look forward to receiving your revised manuscript.

Kind regards,

Ogugua Ndubuisi Okonkwo, M.D.

Academic Editor

PLOS ONE

Journal requirements:   When submitting your revision, we need you to address these additional requirements. 1. Please ensure that your manuscript meets PLOS ONE's style requirements, including those for file naming. The PLOS ONE style templates can be found at https://journals.plos.org/plosone/s/file?id=wjVg/PLOSOne_formatting_sample_main_body.pdf and https://journals.plos.org/plosone/s/file?id=ba62/PLOSOne_formatting_sample_title_authors_affiliations.pdf.

Reviewers' comments:

Reviewer's Responses to Questions

**Comments to the Author**

1. Is the manuscript technically sound, and do the data support the conclusions?

Reviewer #1: Yes

Reviewer #2: Yes

2. Has the statistical analysis been performed appropriately and rigorously? 

Reviewer #1: Yes

Reviewer #2: Yes

3. Have the authors made all data underlying the findings in their manuscript fully available?

Reviewer #1: No

Reviewer #2: Yes

4. Is the manuscript presented in an intelligible fashion and written in standard English?

Reviewer #1: Yes

Reviewer #2: Yes

5. Review Comments to the Author

Reviewer #1: Reviewer Comment

The authors report the Risk factors for hypotony after transconjunctival sutureless vitrectomy. Please address the following questions and comments to improve the manuscript.

The authors should be added the reference and discuss the similarities and differences their study.

Ercalik NY, Tekcan H, Mangan MS, Ozcelik Kose A, Imamoglu S. Analysis of risk factors and ocular hypotony characteristics in choroidal detachment after penetrating glaucoma surgery. Int Ophthalmol. 2022 Nov;42(11):3431-3440. doi: 10.1007/s10792-022-02342-1. Epub 2022 May 20.

Reviewer #2: This manuscript entitled “Risk factors for hypotony after transconjunctival sutureless vitrectomy” presents the risk factors for postoperative hypotony after sutureless small gauge vitrectomy in a very large sample of eyes.

The study has a very large sample of enrolled eyes, many different underlying pathologies and a good methodology regarding statistical analysis and group order by AL. Overall, this data can be very interesting for the community and contributes to our knowledge by suggesting clearly the sutures in high myopic eyes.

However, there are some minor issues, that need to be addressed.

• First of all, please correct the AL by 20-23mm, 23-26mm and ≥26mm. It sounds much more rational, and we can include the ≤ symbol.

• Secondly, can should discuss why one VR surgeon would not suture after silicone oil-removal or after re-vitrectomy? We know that if vitreous base shaving is performed, the residual vitreous is too small to contribute in would sealing. Especially, in case of re-surgery, if the same trocar position is used, the wound healing process is further delayed. Furthermore, eyes after silicone oil removal, show more frequently a postoperative hypotony, even with gaseous tamponades. This was also shown in your cohort. Based on your univariate analysis, you may suggest also suturing not only in high myopic eyes (where it is known that sclera is thinner) but also in re-surgeries and after SOR (silicone oil removal).

• Which was the postoperative therapy? Do you inject steroids subconjuctival or parabulbous at the end of the surgery? Steroids at the end of the surgery are applied very often, together with antibiotics. However although they reduce postoperative inflammation, they prevent quick wound closure. In many cases of TSV reducing steroids during the first day or avoiding steroids at the end of the surgery allows quicker wound healing and restoration of normal pressure within 24h. One risks higher inflammation, however, this can addressed much easier if the pressure recovers, by increasing steroids. Patching the eye is a “questionable” measurement, that should be rather avoided. The reason of hypotony in almost all these TSV cases is the “silent” wound leaking, and not ciliar body insufficiency. Please discuss.

6. PLOS authors have the option to publish the peer review history of their article (what does this mean? ). If published, this will include your full peer review and any attached files.

**Do you want your identity to be public for this peer review?** For information about this choice, including consent withdrawal, please see our Privacy Policy .

Reviewer #1: No

Reviewer #2: **Yes: ** Efstathios Vounotrypidis

---

## [Author Response · Author response to Decision Letter 1]

19 Feb 2025

- The academic editor and reviewers' comments to author (in italics, followed by our responses in regular typeface):

Academic editor:

Dear author, your article will be more relevant if findings are compared with intial or prior work on post operative hypotony after transconjunctival vitrectomy. Also include the possible complications of hypotony in your discussion. Kindly revise your article according to the comments and advice of the reviewers.

Response:

We sincerely appreciate your valuable feedback. Following your guidance, we have thoroughly revised our manuscript based on the reviewers’ suggestions. As recommended, we have incorporated comparisons with prior studies, including discussions on potential complications of hypotony, with particular emphasis on choroidal detachment (CD) and hypotony maculopathy. Furthermore, we have expanded our discussion of additional risk factors, as suggested.

Reviewer #1: Reviewer Comment

The authors report the Risk factors for hypotony after transconjunctival sutureless vitrectomy. Please address the following questions and comments to improve the manuscript.

The authors should be added the reference and discuss the similarities and differences their study.

Ercalik NY, Tekcan H, Mangan MS, Ozcelik Kose A, Imamoglu S. Analysis of risk factors and ocular hypotony characteristics in choroidal detachment after penetrating glaucoma surgery. Int Ophthalmol. 2022 Nov;42(11):3431-3440. doi: 10.1007/s10792-022-02342-1. Epub 2022 May 20.

Response:

We sincerely appreciate your valuable suggestion. As recommended, we have cited the referenced article (Ref #29) and integrated its findings into the Discussion section (P20L311–312, P20L322–324). The study you mentioned focused on risk factors for CD in cases with hypotony (intraocular pressure (IOP) < 9 mmHg); therefore, we have expanded our discussion by incorporating incidence rates of hypotony maculopathy following glaucoma surgery: “In contrast, this condition is more commonly associated with complications after glaucoma surgery, with reported incidence rates of 9.4% after trabeculectomy and 6.0% after Ahmed glaucoma valve implantation in eyes with hypotony (IOP < 9 mmHg) [29]” (P20L322–324).

Reviewer #2: This manuscript entitled “Risk factors for hypotony after transconjunctival sutureless vitrectomy” presents the risk factors for postoperative hypotony after sutureless small gauge vitrectomy in a very large sample of eyes.

The study has a very large sample of enrolled eyes, many different underlying pathologies and a good methodology regarding statistical analysis and group order by AL. Overall, this data can be very interesting for the community and contributes to our knowledge by suggesting clearly the sutures in high myopic eyes.

However, there are some minor issues, that need to be addressed.

• First of all, please correct the AL by 20-23mm, 23-26mm and ≥26mm. It sounds much more rational, and we can include the ≤ symbol.

Response:

We sincerely appreciate your valuable suggestion. We agree that the classification you proposed is more rational and, as recommended, have accordingly revised the axial length categories to '20–23 mm, 23–26 mm, and ≥26 mm'. In line with this revision, we have also updated the Abstract (P2L38), the manuscript (P16L228–234), Figure 2, and its legend.

• Secondly, can should discuss why one VR surgeon would not suture after silicone oil-removal or after re-vitrectomy? We know that if vitreous base shaving is performed, the residual vitreous is too small to contribute in would sealing. Especially, in case of re-surgery, if the same trocar position is used, the wound healing process is further delayed. Furthermore, eyes after silicone oil removal, show more frequently a postoperative hypotony, even with gaseous tamponades. This was also shown in your cohort. Based on your univariate analysis, you may suggest also suturing not only in high myopic eyes (where it is known that sclera is thinner) but also in re-surgeries and after SOR (silicone oil removal).

Response:

We sincerely appreciate your valuable insights. The increased risk of postoperative hypotony in eyes undergoing silicone oil removal (SOR) or re-vitrectomy has been well documented, and we fully agree with the mechanisms you highlighted. However, the VR surgeon in our study did not deliberately perform all cases without suturing. As described in the Materials and methods section, sutures were placed when an obvious wound closure failure was observed, and such cases were excluded from the analysis. Thus, the cases included in our dataset were expected to maintain stable IOP postoperatively, making the hypotony observed the following day unexpected.

Nevertheless, our results further confirm that, despite normal IOP at the end of surgery, many cases with SOR or re-vitrectomy may develop postoperative hypotony due to “silent” wound leakage, as discussed in the following comment. To mitigate this risk, suturing should be considered proactively in modern small-gauge vitrectomy, including 27G transconjunctival sutureless vitrectomy (TSV), rather than adhering strictly to its benefits. Although SOR and re-vitrectomy were not statistically significant risk factors in our multivariate analysis, their relevance has been emphasized in previous studies and was evident in our univariate results, warranting further discussion.

Accordingly, we have incorporated the following paragraph into the Discussion section: “In addition to high myopia and no tamponade usage, previous vitrectomy and SO removal have been identified as significant risk factors for hypotony following vitrectomy [14,16,17], though this association was only observed in our univariate analysis. If vitreous base shaving is performed, the residual vitreous may become insufficient for wound sealing in subsequent vitrectomy procedures [14,16]. Additionally, reusing the same trocar position may delay the wound healing process due to scleral scarring or thinning [14,16]. In cases of SO removal, the presence of residual SO within the wound may impair adequate self-sealing, and these eyes have typically undergone prior vitrectomy [17]. Although some surgeons in our study opted not to suture in these cases after confirming IOP, there was no deliberate attempt to avoid suturing in all cases. The primary mechanism of hypotony in almost all TSV cases is likely postoperative “silent” wound leakage, except for the ciliary body dysfunction due to uveitis [20]. Given these mechanisms, proactive suturing should still be considered in cases with these risk factors to prevent such leakage, even in modern small-gauge vitrectomy.” (P18L281–P19L293).

We sincerely appreciate your thoughtful comments, which have significantly contributed to refining our discussion.

• Which was the postoperative therapy? Do you inject steroids subconjuctival or parabulbous at the end of the surgery? Steroids at the end of the surgery are applied very often, together with antibiotics. However although they reduce postoperative inflammation, they prevent quick wound closure. In many cases of TSV reducing steroids during the first day or avoiding steroids at the end of the surgery allows quicker wound healing and restoration of normal pressure within 24h. One risks higher inflammation, however, this can addressed much easier if the pressure recovers, by increasing steroids. Patching the eye is a “questionable” measurement, that should be rather avoided. The reason of hypotony in almost all these TSV cases is the “silent” wound leaking, and not ciliar body insufficiency. Please discuss.

Response:

We sincerely appreciate your valuable comments and insights. In response to your suggestion, we have added a description of the subconjunctival steroid injection at the end of the surgery to the Surgical Procedure section in the Materials and methods: “At the end of the surgery, 0.1 mL of dexamethasone was injected into the subconjunctiva.” (P7L139–140). We acknowledge that steroids may delay wound healing and contribute to postoperative hypotony. A previous study identified intraoperative triamcinolone acetonide (TA) use as a significant risk factor for hypotony in univariate analysis (Ref #14, Bamonte G, et al. Am J Ophthalmol. 2011). Interestingly, our study yielded an unexpected finding: the frequency of TA use was actually lower in the hypotony group (88.5%) compared to the non-hypotony group (94.4%, p = 0.010, Table 3). However, this finding alone does not completely exclude the potential influence of steroids on postoperative hypotony. To our knowledge, Ref #14 remains the only study that has explicitly examined the relationship between steroid use and post-vitrectomy hypotony.

As you rightly pointed out, higher inflammation can be easily managed with medication, whereas increasing IOP from a hypotonic state is far more challenging. Therefore, the impact of intraoperative triamcinolone and subconjunctival steroid injections in vitrectomy should be further investigated in future studies.

The eye patch was not intended to prevent hypotony but rather to protect the eye from physical contact, such as rubbing by the patient, during the first postoperative night.

Finally, we fully agree with your statement that silent wound leakage, rather than ciliary body dysfunction, is the primary cause of hypotony after TSV.

We sincerely appreciate your thoughtful suggestions, which have significantly contributed to refining our discussion.

---

## [Decision Letter · Decision Letter 1]

2 Mar 2025

Risk factors for hypotony after transconjunctival sutureless vitrectomy

PONE-D-25-02954R1

Dear Dr. Ando,

We’re pleased to inform you that your manuscript has been judged scientifically suitable for publication and will be formally accepted for publication once it meets all outstanding technical requirements.

Kind regards,

Ogugua Ndubuisi Okonkwo, M.D.

Academic Editor

PLOS ONE

Additional Editor Comments (optional):

The authors have addressed satisfactorily all the concerns and comments raised in the initial round of review.

Reviewers' comments:

Reviewer's Responses to Questions

**Comments to the Author**

1. If the authors have adequately addressed your comments raised in a previous round of review and you feel that this manuscript is now acceptable for publication, you may indicate that here to bypass the “Comments to the Author” section, enter your conflict of interest statement in the “Confidential to Editor” section, and submit your "Accept" recommendation.

Reviewer #1: All comments have been addressed

Reviewer #2: All comments have been addressed

2. Is the manuscript technically sound, and do the data support the conclusions?

Reviewer #1: Yes

Reviewer #2: Yes

3. Has the statistical analysis been performed appropriately and rigorously? 

Reviewer #1: Yes

Reviewer #2: Yes

4. Have the authors made all data underlying the findings in their manuscript fully available?

Reviewer #1: No

Reviewer #2: No

5. Is the manuscript presented in an intelligible fashion and written in standard English?

Reviewer #1: Yes

Reviewer #2: Yes

6. Review Comments to the Author

Reviewer #1: nice revision, Risk factors for hypotony after transconjunctival sutureless vitrectomy about hypotony

Reviewer #2: The authors have addressed the reviewers comments adequately and the manuscript is now suitable for publication.

7. PLOS authors have the option to publish the peer review history of their article (what does this mean? ). If published, this will include your full peer review and any attached files.

**Do you want your identity to be public for this peer review?** For information about this choice, including consent withdrawal, please see our Privacy Policy .

Reviewer #1: No

Reviewer #2: No

---

## [Editor Report · Acceptance letter]

PONE-D-25-02954R1

PLOS ONE

Dear Dr. Ando,

I'm pleased to inform you that your manuscript has been deemed suitable for publication in PLOS ONE. Congratulations! Your manuscript is now being handed over to our production team.

Kind regards,

on behalf of

Dr. Ogugua Ndubuisi Okonkwo

Academic Editor

PLOS ONE